

Towards resolving the complex paramagnetic NMR spectrum of small laccase:
Assignments of resonances to residue specific nuclei

Rubin Dasgupta, Karthick B.S.S. Gupta, Huub J.M. de Groot, Marcellus Ubbink*

Leiden Institute of Chemistry, University of Leiden, Gorlaeus Laboratory, Einsteinweg 55, 2333 CC, Leiden, The Netherlands.

*Correspondence to*: Marcellus Ubbink (m.ubbink@chem.leidenuniv.nl)

**Abstract**

Laccases efficiently reduce dioxygen to water in an active site containing a tri-nuclear copper centre (TNC). One reason for its efficiency in catalysis of this complex reaction can be the presence of mobility of active site residues. To probe mobility, NMR spectroscopy is highly suitable. However, several factors complicate the assignment of resonances to active site nuclei in laccases. The paramagnetic nature causes large shifts and line broadening. Furthermore, the presence of slow chemical exchange

processes of the imidazole rings of copper ligands result in peak doubling. A third complicating factor is that the enzyme occurs in two states, the native intermediate (NI) and resting oxidized (RO) states, with different paramagnetic properties. The present study aims at resolving the complex paramagnetic NMR spectra of the TNC of *Streptomyces coelicolor* small laccase (SLAC). With a combination of paramagnetically tailored NMR experiments, all eight His N$\delta$1 and H$\delta$1 resonances for the NI state are

identified, as well as His H$\beta$ protons for the RO state. With the help of second shell mutagenesis, selective resonances are tentatively assigned to the T2 histidines. This study demonstrates approaches that can be used for sequence specific assignment of the paramagnetic NMR spectra of ligands in the TNC that ultimately may lead to a description of the underlying motions.

**Keywords**: Paramagnetic NMR spectroscopy, WEFT, $^1$H-$^{15}$N HMQC, small laccase, tri-nuclear copper centre.

## 1. Introduction

Multicopper oxidases (MCOs) oxidize a wide variety of substrates at their type 1 (T1) site and

catalyse the 4-electron reduction of molecular oxygen to water at the tri-nuclear copper centre (TNC). The TNC consists of a type 2 (T2) copper site and a binuclear type 3 (T3) copper site. Based on crystallographic, spectroscopic and theoretical studies, the present model of the oxygen reduction mechanism by the TNC is shown in Scheme 1 (Gupta et al., 2012; Heppner et al., 2014; Quintanar et al., 2005b; Tepper et al., 2009; Yoon and Solomon, 2007). The two-domain small laccase from

*Streptomyces coelicolor* (SLAC) has been reported to involve the formation of a tyrosine radical (Tyr108$^\bullet$) near the T2 site during the peroxide intermediate (PI) to native intermediate (NI) conversion (Gupta et al., 2012; Tepper et al., 2009). This radical has been suggested to act as protection against the reactive oxygen species (ROS) that can be formed due to the long-lived peroxide intermediate state (Gupta et al., 2012; Kielb et al., 2020). The tyrosyl radical was shown to be reduced by the protein



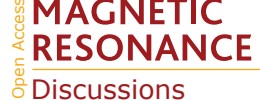

environment via tryptophan and tyrosine residues around the T2 site (Kielb et al., 2020). A similar role was proposed for Tyr107 in human ceruloplasmin (hCp). hCp is a ferroxidase critical for iron homeostasis. It oxidizes $Fe^{2+}$ to $Fe^{3+}$ for iron transport. In serum the hCp is active under low $Fe^{2+}$ and high $O_2$ concentration. This leads to a partially reduced intermediate that can form ROS. The tyrosine radical protects the protein from this partially reduced state (Tian et al., 2020).


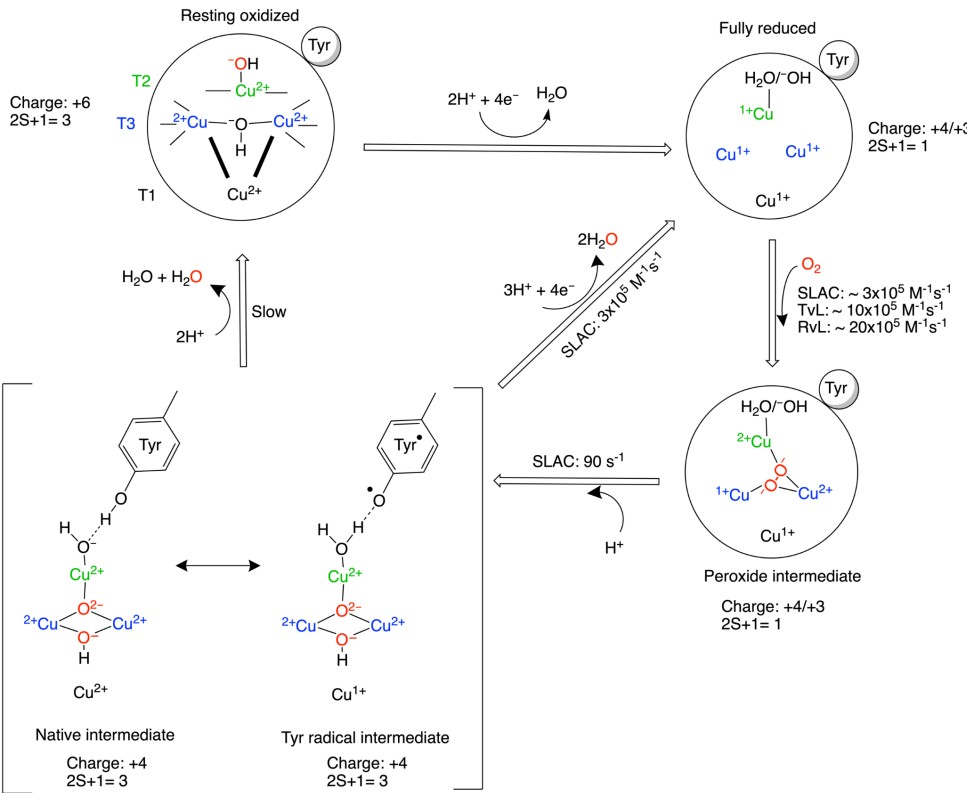

**Scheme 1**: Reaction mechanism of the oxygen reduction reaction in SLAC. The coordination of the coppers in the TNC is shown in the resting oxidized state. The T3 coppers (blue) are coordinated to
three histidine Nε2 atoms and the hydroxyl group, whereas the T2 copper (green) is coordinated to two histidine Nε2 atoms and a water/hydroxide ligand. The charges and the spin multiplicities (2S+1, where S is the total spin in the system) are shown based on the literature and reflect ground states (Gupta et al., 2012; Heppner et al., 2014; Quintanar et al., 2005b; Yoon and Solomon, 2007). The rates for oxygen binding are shown for laccases from several organisms, SLAC from *S. coelicolor*, TvL from *Trametes*
*versicolor* laccase and RvL for *Rhus vernicifera* laccase (Heppner et al., 2014). An intermediate for SLAC is shown with the Tyr• radical (Gupta et al., 2012; Tepper et al., 2009). This intermediate has only been reported for SLAC and hCp (Tian et al., 2020).



Although the reaction mechanism of laccase is well characterized, information about motions around the TNC is limited. Such mobility in active sites was shown to promote resonant transfer from reactant into product by vibronic coherence during the electron and proton transfer for high yield (Dorner et al., 2012; Lloyd, 2011; Purchase and de Groot, 2015; Zhang and Wang, 2016). Ultimately, characterisation such motions at the TNC of laccase can help in designing a functional framework for understanding the natural process and the *de novo* design of efficient bioinspired catalysts. Three or more independent chemical exchange processes, tentatively assigned to the coordinating histidine residues at the TNC were reported using paramagnetic NMR spectroscopy on the T1 copper depleted variant of SLAC, SLAC-T1D (Dasgupta et al., 2020). However, further characterisation of motions requires assignments of the NMR resonances very near to the TNC. The paramagnetic nature of the coppers causes broadening and chemical shifts outside of the diamagnetic envelope, making it impossible to employ standard multidimensional protein assignment experiments. Assignment is further complicated by two reasons. SLAC spectra are a mixture of the RO and NI states (Scheme 1) (Machczynski and Babicz, 2016). In the RO state the T2 $Cu^{2+}$ is isolated, causing broadening of the signals of nearby proton spins beyond detection. The two coppers in the T3 site are antiferromagnetically coupled, with a low-lying triplet state that is populated at room temperature, causing paramagnetically shifted (in the range of 12 - 22 ppm), detectable resonances of nearby protons. In the NI state all coppers are coupled, resulting in a frustrated spin system, with strongly shifted (> 22 ppm), but observable resonances (Zaballa et al., 2010). The second cause of complexity is that the mentioned exchange processes of the coordinating histidine residues results in peak doublings, because the exchange rates are in the slow exchange regime relative to the resonance frequency differences. In this study, we aimed to resolve further this complicated paramagnetic NMR spectrum. Using differently labelled samples and tailored HMQC experiments, the presence of all eight histidine ligand N$\delta$1 and H$\delta$1 resonances in the NI state could be established. The first studies of the RO state identified resonances as histidine H$\delta$1 or H$\beta$ protons and a second coordination shell mutant allowed for the first residue and sequence specific assignment. The study demonstrates approaches that may lead to further assignments and ultimately a full description of motions in laccase active sites.

## 2. Results and discussion

*2.1. Identification of nitrogen attached protons in the NI state.* The Fermi contact shifted [1]H resonances > 21 ppm from the TNC of SLAC-T1D that exhibited exchange processes are probably from the histidine H$\delta$1 protons (Dasgupta et al., 2020). To verify this assignment, a paramagnetically tailored [1]H-[15]N HMQC experiment (Figure S2) was performed on a SLAC-T1D sample that was specifically labelled with [15]N histidine in a perdeuterated, back-exchanged environment. The evolution period was shortened to 500 $\mu$s, balancing the time required for formation of antiphase magnetisation and paramagnetic relaxation, to optimize S/N ratio for most of the resonances. A total of 10 resonances (3, 4, 5, 6, 9, 11, 12, 13, 14/15, 16, see Figure 1b) were observed at [1]H chemical shifts of > 21 ppm. Resonance 7, 8 and 10 were not observed in this experiment, which is consistent with their assignment

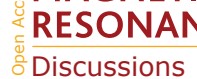



to carbon attached protons (Dasgupta et al., 2020). These results show unequivocally that the HMQC resonances derive from the Hδ1 protons of the coordinating histidine residues of the TNC, because only these protons are nitrogen attached and close enough to experience such large paramagnetic shifts. The three pairs or resonances representing exchange processes (3-5, 9-11 and 13-12) are thus also from Hδ1 protons, in line with the suggested histidine ring motion being the involved chemical exchange process. The HMQC spectrum of uniformly 15N labelled SLAC-T1D is similar to the 15N-His specifically labelled SLAC-T1D sample in a perdeuterated back-exchanged environment (data not shown for the 1H resonances > 21 ppm but shown for the region 12 to 21 ppm, see Figure 2b).

SLAC-T1D is predominantly in the NI state, in which the T2 and the T3 sites are coupled, increasing the electronic relaxation rates of the unpaired electrons and thus reducing the paramagnetic relaxation rates of the nuclear spins. Therefore, it is expected that all eight ligand histidine residues are observable. In the 1H-15N HMQC ten resonances are seen, among which three undergo chemical exchange resulting in the observation of seven Nδ1-Hδ1 groups. Resonance 17 and 18 have exchange cross-peaks with resonance 15/14 and 16, respectively at high temperatures (303 K and 308 K) and a short mixing time in an EXSY/NOESY experiment (1 and 2 ms) (Dasgupta et al., 2020). At temperatures of 298 K and higher, resonances 14 and 15 overlap (Figure 1a) (Dasgupta et al., 2020). Resonance 16 and 18 thus form a fourth exchange pair and the eighth histidine Nδ1-Hδ1 group can be attributed to the exchange pair of resonance 17 with either 14 or 15 (Dasgupta et al., 2020). Due to the overlapping of resonance 14 and 15 at 298 K, they are not observed distinctly in 1H-15N HMQC spectra (Figure 1b). In conclusion, all the eight Hδ1 from the coordinating histidines of the TNC in SLAC-T1D for the NI state are identified in the spectral region > 21 ppm and five of them show peak doubling due to slow exchange.

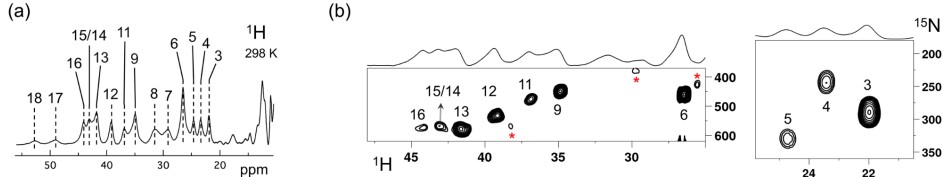

**Figure 1**. SLAC-T1D NMR spectra at 298 K. (a) 1D 1H WEFT spectrum of SLAC-T1D and (b) 15N-1H HMQC spectra of 15N-His perdeuterated SLAC-T1D in a back-exchanged environment. The numbering is adopted from (Dasgupta et al., 2020). Noise peaks in the spectrum are marked with a red asterisk. The 1D 1H WEFT spectrum is shown above the HMQC spectrum.

*2.2. Analysis of the RO state.* Machczynski *et al*. reported that the signals in the spectral region between 12 to 21 ppm derive from the RO state, whereas the resonances > 21 ppm are attributed to the NI state (Machczynski and Babicz, 2016). In the RO state, the T2 copper is decoupled from the T3 site, resulting in a decrease of its electronic relaxation rate (Bertini et al., 2017). This effect broadens the resonances of nearby proton spins beyond detection for the T2 site ligands. In the RO state, the T3 coppers are antiferromagnetically coupled and thus diamagnetic at low temperature (Bertini et al., 2017). At ambient temperature, the low-lying state with S = 1 is populated, resulting in paramagnetic





shifts of the ligand protons (Bertini et al., 2017). The strong coupling via a hydroxyl moeity of the electron
spins causes fast electronic relaxation and thus observable nuclear resonances for T3 ligands. T3 site
ligands usually exhibit an anti-Curie behavior, i.e. the chemical shift increases with an increase in
temperature (Bertini et al., 2017, 1993; Bubacco et al., 2000; Tepper et al., 2006).

All the resonances in the 12 to 21 ppm region of SLAC-T1D in an $^1$H-$^1$H EXSY/NOESY
spectrum display anti-Curie behavior, suggesting that indeed they derive from histidine protons of the T3
site (Figure 2). Comparing the $^1$H-$^{15}$N HMQC and the $^1$H-$^1$H EXSY/NOESY of the $^{15}$N uniformly labelled
sample in this region, resonances a1, a2, b2, c1, c2, d2, x1, x2, y, z and w are nitrogen linked protons
(Figure 2). The RO state is the minor state in SLAC-T1D, so the S/N ratio for the HMQC resonances is
low. For comparison, resonance 3, which belongs to the NI region of the spectrum (Figure 2b) is shown
as well. $^1$H resonances e1 and e2 could not be assigned to either carbon or nitrogen linked protons due
to their low S/N ratio.

Using a two-metal centre model to calculate the singlet-triplet energy gap (2J) from the
temperature dependence of the chemical shifts (equation S1), a 2J value of 600 +/- 20 cm$^{-1}$ was
obtained, within the range of the previous reported values (550 to 620 cm$^{-1}$) for the RO state of laccase
(Figure 2c) (Battistuzzi et al., 2003; Machczynski and Babicz, 2016; Quintanar et al., 2005b). It was
assumed that resonances a1, a2, b2, c1, c2, d2 (only isolated resonances were selected) are the Fermi
contact shifted resonances of the Hδ1 of the coordinating histidine residues at the T3 site in the RO
state (Figure 2d), as supported by their presence in the HMQC spectrum (Figure 2b). The diamagnetic
chemical shift for these resonances was set to 9.5 ppm (BMRB average for histidine ring Hδ1) (Zaballa
et al., 2010). To establish the diamagnetic chemical shifts of resonances b1 and d1, which are not
nitrogen attached, the 2J coupling strength was then fixed to 600 cm$^{-1}$ and the diamagnetic chemical
shift was fitted and found to be 3.0 +/- 0.5 ppm. This value strongly suggests that these resonances are
from the β protons of coordinating histidines (BMRB average for histidine Hβ is 3.1 ppm).

Since the temperature dependence of the cross peak intensities as measured by their peak
volume did not show a conclusive increasing trend with increase of temperature, we assumed them to
be NOE rather than EXSY derived cross-peaks (Dasgupta et al., 2020). Therefore, the cross peaks of
resonances b1-b2 and d1-d2 can be attributed to a NOE between the Hδ1 and Hβ proton of a histidine
ligand. The cross-peaks between c1-c2 and a1-a2 appear to be NOE signals from nitrogen attached
protons (Figure 2). The Hδ1 protons of the different histidine residues are not near, so it remains unclear
from which spins these peaks derive. For the resonances x1, x2, y, z and w (Figure 2a) the analysis of
the temperature dependence of the chemical shift was not possible due to the overlap.



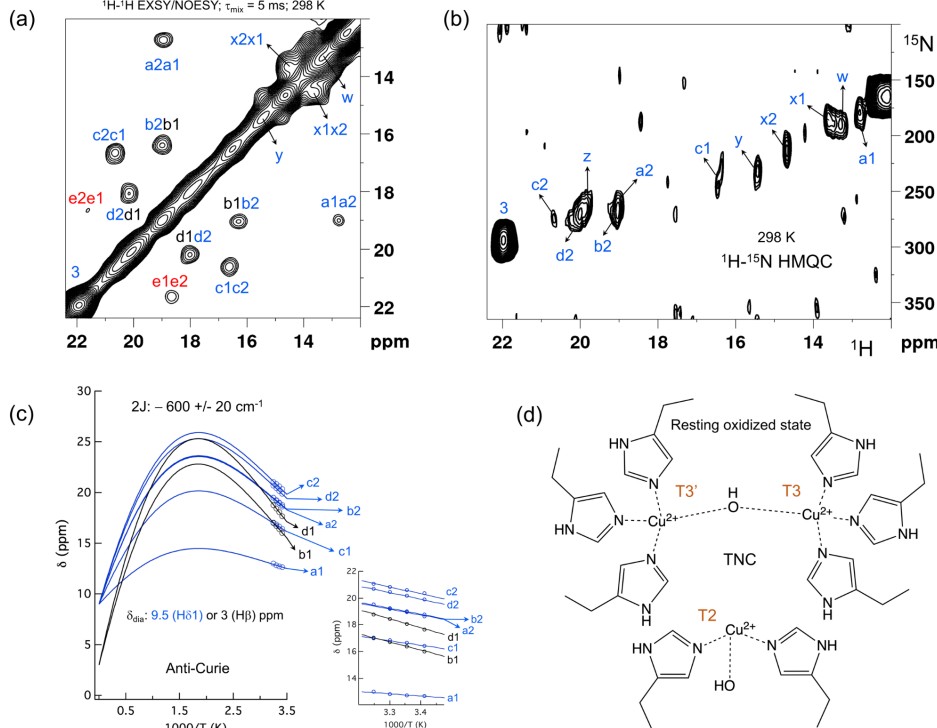

**Figure 2**. The spectral region of the RO state. (a) $^1H$-$^1H$ EXSY/NOESY spectra of SLAC-T1D for the region between 12 to 22 ppm; (b) $^1H$-$^{15}N$ HMQC spectra of the $^{15}N$ uniformly labelled SLAC-T1D (12 to 22 ppm in the $^1H$ dimension); (c) Temperature dependence of the chemical shift for the resonances a1, a2, b1, b2, c1, c2, d1 and d2, fitted to the two-metal center model (equation S1). The inset shows the enlarged region of the fit to the data; (d) Schematic representation of the resting oxidized state of the TNC where the T3 and T2 coppers are marked. The resonances marked in blue are for the histidine Hδ1 nuclei. Resonances in black are for carbon attached protons. Resonances in red in panel a could not be assigned to either nitrogen linked or carbon linked protons due to a low S/N ratio.

*2.3. Second shell mutagenesis to assist assignments.* To aid in the assignment of the paramagnetic spectrum, mutagenesis could be employed. However, mutation of histidine ligands is expected to result in loss of copper or at least severe redistribution of unpaired electron density, changing the chemical shifts of all paramagnetically shifted protons. In contrast, mutations in the second coordination sphere, of residues that interact with the coordinating ligands, may have moderate effects on the electron spin density distribution. One such mutant, Y108F, has been reported before (Gupta et al., 2012). Tyr108 interacts with the TNC in two ways, with the T2 site through the water/hydroxide ligand and with the T3 ligand His104 through the hydrogen bonding network involving Asp259 (Figure S3a). Asp259 is conserved in all laccases, whereas Tyr108 is conserved in the two-domain laccases (Figure S3b). Asp259 has been reported to play a role in modulating the proton relay during the oxygen reduction reaction (Quintanar et al., 2005a, p.94) and it may also stabilize the Tyr108-TNC interaction.

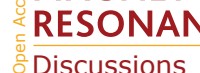

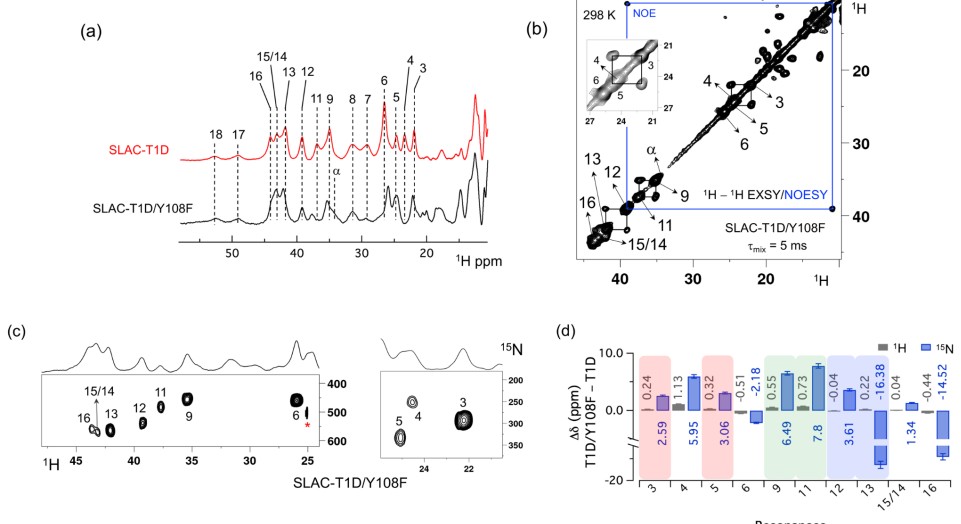

**Figure 3**. Spectra of SLAC-T1D/Y108F. (a) Comparison between 1D $^1$H WEFT spectrum of SLAC-T1D (red) and SLAC-T1D/Y108F (black). The numbering is shown for SLAC-T1D and is adopted from (Dasgupta et al., 2020); (b) $^1$H-$^1$H EXSY/NOESY of SLAC-T1D/Y108F at 298 K with mixing time of 5 ms. NOE cross-peaks are connected with a blue rectangle. The remaining cross-peaks are exchange peaks. This distinction is based on the temperature dependent profile of the integral volume of the cross peaks as explained in (Dasgupta et al., 2020). The inset shows that the exchange cross peaks are between 3 and 5. Resonance 4 is partly overlapping with 5; (c) $^1$H-$^{15}$N HMQC spectra of $^{15}$N uniformly labelled SLAC-T1D/Y108F; (d) The chemical shift changes ($\Delta\delta$) between SLAC-T1D/Y108F and SLAC-T1D for the $^1$H (black) and $^{15}$N (blue). The error bars represent the standard deviation in the determination of the chemical shift. The three pairs of resonances displaying chemical exchange are highlighted by equal background colours. Positive (negative) values represent shift to the downfield (upfield) ppm for SLAC-T1D/Y108F.

The 1D $^1$H WEFT (Bertini et al., 1993; Patt and Sykes, 1972) spectrum of SLAC-T1D/Y108F is similar to that of SLAC-T1D, suggesting that the variant SLAC is also predominantly in the NI state (Figure 2a). Some changes in the chemical shift are present. Due to the Y108F mutation many of the $^1$H resonances > 22 ppm are downfield shifted. Resonance 6 and 16 are upfield shifted and resonance 7, 8, 17 and 18 show no chemical shift change compared to SLAC-T1D (Figure 3 and Table S2). Also, a new resonance $\alpha$ is observed. The HMQC spectrum in the region > 22 ppm of the $^1$H is very similar to that of SLAC-T1D, in agreement with the $^1$H WEFT spectrum (Figure 3). Most of the $^{15}$N resonances (3, 4, 5, 9, 11, 12 and 15) are downfield shifted except resonances 6, 13 and 16, which are upfield shifted (Figure 3 and Table S2). The three independent chemical exchange processes that were reported for the TNC of SLAC-T1D involving resonance pairs of $3-5$, $9-11$ and $13-12$ (Dasgupta et al., 2020) are conserved and the rates are not affected by the Y108F mutation (Table S1, Figure 3b and Figure S1c), suggesting that the phenolic $-$OH group of Y108 is not involved in the chemical exchange process. The chemical shift changes show that the two states represented by $3-5$ and $9-11$, respectively are affected similarly by the Y108F mutation (Figure 3d). In contrast, the two states


represented by the resonance pair 13 − 12 are affected differently, because the nitrogen chemical shift is downfield shifted for resonance 12 and to upfield shifted for resonance 13 (Figure 3d).

It is proposed that resonances 13 and 16, which are most affected by the Y108F mutation (Figure 3d), are from the histidine ligands of the T2 copper. Due to the proximity of the T2 copper and strong hydrogen bond with a water or hydroxide ligand, the electron spin density can be expected to be delocalized to the tyrosine ring. The loss of the hydrogen bond between the phenolic -OH group of

Tyr108 and the water/hydroxide ligand of the T2 copper can result in redistribution of the electron spin density on the coordinating histidine ligands. Figure 3d shows that the $N\delta 1$ of the resonances 13 and 16 have the highest chemical shift perturbation of ~ -16 and -14 ppm respectively. Interestingly, resonance 13 is in an exchange process with resonance 12 (Figure 3b) (Dasgupta et al., 2020) and for the latter resonance the $N\delta 1$ exhibits a downfield shift due to the Y108F mutation. In the crystal structure

3cg8 (resolution 2.63 Å), the $N\delta 1$ of His102 from the T2 site can have two hydrogen bonding partners, carbonyl oxygen of Asp113 and a water molecule (Figure 4a). Modelling the protons and changing the $\chi 2$ dihedral angle of His102 to -152º and -94º, hydrogen bonds can be formed between $H\delta 1$ — Asp113 CO and $H\delta 1$ — $H_2O$ respectively. The $\chi 2$ dihedral change does not break the coordination of His102 $N\epsilon 2$ to the copper (Figure 4b and 4c) and is within the allowed range (-90º to -170º) (Dasgupta et al.,

2020). This shows that there can be a conformational exchange of His102 between two states with a hydrogen bond between $H\delta 1$ and either Asp113 CO or the nearby $H_2O$ molecule. The second shell mutation of Y108F suggests that the exchanging resonances 13 and 12 are from a $H\delta 1$ nucleus of one of the two T2 copper histidine ligands. Thus, it is proposed that resonance 13 and 12 are from His102 $H\delta 1$ for which the ring exchanges between the two states shown in panels Figure 4b and 4c.

Consequently, resonance 16 can be tentatively assigned to the other T2 copper ligand, His234, being also strongly affected by the Y108F mutation. It does not exhibit chemical exchange at temperatures ≤ 298 K, in agreement with having a single, hydrogen bond with Asp259 CO (Figure 4a). At higher temperatures ( ≥ 303 K) however, exchange with resonance 18 is observed. Whereas the 12/13 pair of resonances shows a difference of less than 3 ppm (Dasgupta et al., 2020) and similar linewidth for both

signals, the 16/18 pair shows almost 9 ppm difference in chemical shift and resonance 18 is much broader, indicating a more drastic change in spin density on the proton. In combination with the observation that there are no other hydrogen bond acceptors in the proximity, this suggests that resonance 18 represents the His234 $H\delta 1$ in a state in which the hydrogen bond to Asp259 is broken. In such a state the proton would be prone to exchange with bulk water protons but the TNC is very

buried, preventing rapid exchange. Similar situations as for His102 are observed for other histidine ligands in the TNC (Table S4). For example, in the crystal structure 6s0o (resolution 1.8 Å) (Gabdulkhakov et al., 2019) $N\delta 1$ of His237 can form a hydrogen bond with Asp114 $O\delta 1$ or water O540, depending on rotation around $\chi 2$ (Figure S5). In the crystal structure 3cg8 the equivalent Asp113 $O\delta 1$ is moved away from the $N\delta 1$ and therefore could not form a hydrogen bond (Figure S5a). Such

exchange processes may well represent the resonances pair 3-5 and 9-11. Second-shell mutations around the respective histidine residues can help to confirm this hypothesis.

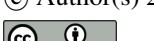

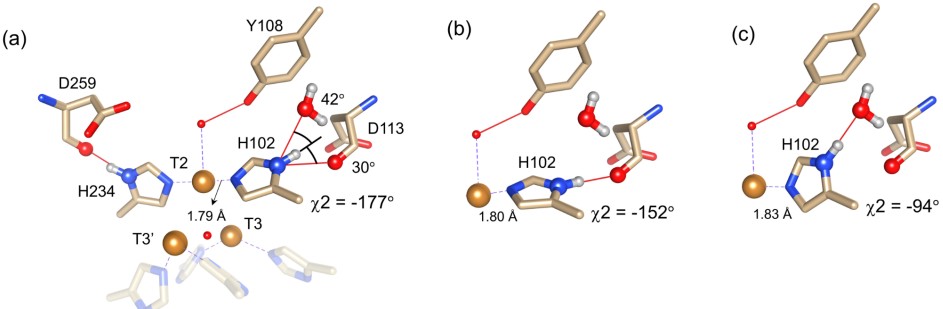

**Figure 4**. Alternative hydrogen bond acceptors for His102. (a) T2 site histidine ligands showing the hydrogen bonds for the Nδ1-Hδ1 groups. Protons were modelled using the algorithm as implemented in UCSF Chimera (Pettersen et al., 2004). His104 and H236 from the T3' and T3 sites, respectively, are omitted for clarity. Hydrogen bonds are shown as red lines. The χ2 dihedral angle and distance between His102 Nε2 and the T2 copper are indicated. Also, the values for the angles [Asp113 CO – His102 Nδ1 – His102 Hδ1] and [water O628 – His102 Nδ1 – His102 Hδ1] are indicated. Ring rotation brings the Hδ1 in optimal position for hydrogen bond formation with either the Asp113 CO (b) or the water (c). The new χ2 dihedral angles and the corresponding His102 Nε2 — T2 copper distances are indicated.

The temperature dependence of Hδ1 resonances is also affected by the Y108F mutation (Figure 5). While the resonances that show clear Curie behaviour in SLAC-T1D also do so in the Y108F mutant, resonances that show anti-Curie or non-Curie behaviour tend more to Curie like behaviour, e.g. resonances 3, 6, 7 and 8. The overall increase in the Curie-like behaviour for the Y108F mutant compared to that of SLAC-T1D, can be due to a change in the geometry of the TNC (Solomon et al., 2008) caused by the loss of the hydrogen bond between the Tyr108 the water/hydroxide.

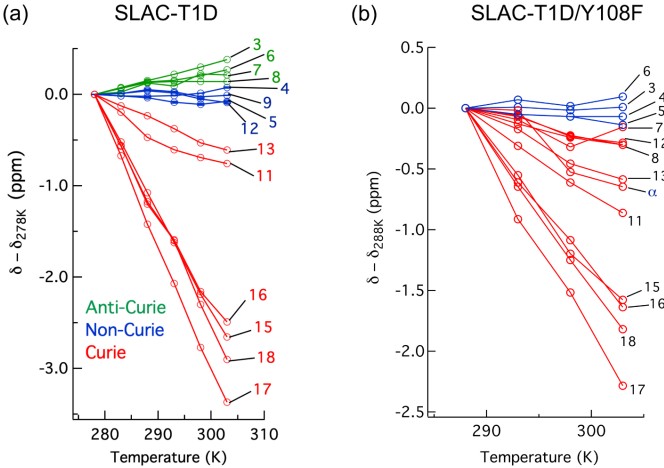



**Figure 5**. Change in $^1$H chemical shifts for (a) SLAC-T1D with temperature relative to 278 K and (b) SLAC-T1D/Y108F with temperature relative to 288 K. Anti-Curie, non-Curie and Curie behaviour are shown in green, blue and red, respectively.

Slight chemical shift changes are also present for the $^1$H resonances between 10 and 20 ppm in the spectrum of SLAC-T1D/Y108F relative to that of SLAC-T1D (Figure S4). The $^1$H-$^1$H EXSY/NOESY spectrum shows six cross-peaks (a to f), caused by 12 diagonal signals (Figure S4). Among these, a1, b1, c1, c2, d1 and e1 are downfield shifted for the mutant, whereas a2, b2, d2 and e2 are upfield shifted (Figure S4b).

In summary, the Y108F mutation leads to the following tentative assignment of the resonances:
13 and 12 to His102 and 16 to His234 of the T2 site, with 13 and 12 being in chemical exchange.

### 3. Conclusion

The SLAC-T1D comprises resonances from the NI and RO states, in which the RO state is the minor
state (Machczynski and Babicz, 2016). Using differently labelled samples and a paramagnetically tailored $^1$H-$^{15}$N HMQC experiment, all NI resonances of the N$\delta$1-H$\delta$1 groups of the eight coordinating histidine residues in the TNC were accounted for (Figure 1). The HMQC spectra also included the resonances that are in chemical exchange, consistent with the histidine ring motions being responsible for this phenomenon (Dasgupta et al., 2020). NOE cross-peaks for the RO state revealed resonances
of H$\beta$ protons of the coordinating histidine residues of the T3 site (Figure 2). The second shell mutation of Y108F of SLAC-T1D aided in the tentative assignment of the resonances 13 and 12 to His102 and 16 and 18 to His234 of the T2 site (Figures 3 and 4). This report shows the first sequential assignment of the resonance to a coordinating histidine. More studies using second shell residue mutagenesis can help to provide a full sequence specific assignment, which is a prelude to a better understanding of the
motions important for the catalytic mechanism.

### 4. Competing interest

The authors declare that they have no conflict of interest.

### 5. Authors contribution

MU and HJMG conceived the project and obtained the required funding, KBSSG and RD optimized the NMR pulse sequence, RD performed the experiment, RD and MU analysed the data, all authors contributed in finalizing the manuscript.

### 6. Acknowledgement

The study was supported by Netherlands Magnetic Resonance Research School (NWO-BOO 022.005.029). We thank Anneloes Blok for performing SEC-MALS on the protein samples.



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
