# Peer review of "Towards resolving the complex paramagnetic NMR spectrum of small laccase: Assignments of resonances to residue specific nuclei"

_Magnetic Resonance, 2020_

## Referee Comment (RC1) · Anonymous Referee #1 · 15 Dec 2020

The manuscript describes the assignment of the paramagnetic NMR spectrum of a small laccase protein, containing a trinuclear copper center. The spectrum has been previously reported, but, In this study, the authors perform the assignment of all hyperfine shifted resonances and succeed in rationalizing the patterns of signals due to chemical exchange processes and to the occurrence of two different redox states: the resting oxidized state and the intermediate state, each of them characterized by peculiar and distinctive features in terms of electron relaxation rates and contact chemical shift ranges. Finally, due to the chemical shift differences observed between the WT protein and Y108F variant, the authors propose the presence of a conformational exchange, involving hydrogen bonds of His 102, which would justify the observed chemical exchange phenomena, and identify at the molecular level the nature of the pre-viously evoked mobility of active site residues. The manuscript is well written, with a very high level of technical skills and a clear description of the results. I recommend its publication, provided the authors addressed a few comments listed below. Also, I would suggest to summarize the assignment of signals as well as the correlations aris-ing from chemical exchange and from the two states, the NI state and the RO state, in a single Table. Comments: Abstract, line 21. Although it is explained in the introduction section, the expression "T2 histidines" in the abstract is unclear. Please replace it with "histidine residues belonging to the type-2 copper site". Abstract, lines 21-24. The last sentence of the abstract is not clear. Perhaps you might replace it with "This study demonstrates the utility of the approaches used for the sequence specific assignment of the paramagnetic NMR spectra of ligands in the TNC that ultimately may lead to a description of the underlying motions"

line 49, line 69, etc: replace "coppers" with "copper ions" [please spell check throughout the text]

lines 84-85: same as lines 21-24.

Line 96: The evolution period was shortened to 500 $\mu$s, balancing the time required for formation of antiphase magnetization and paramagnetic relaxation, to optimize S/N ratio for most of the resonances. Actually, previous works on this aspect (such as "I. Gelis, et al; A simple protocol to study blue copper proteins by NMR, Eur. J. Biochem. 270, 600–609, 2003" or "S. Ciofi-Baffoni et al; The IR-15N-HSQC-AP experiment: a new tool for NMR spectroscopy of paramagnetic molecules, Biomol NMR, 58:123-128, 2014") should be quoted here.

Lines 97-100: The identification and numbering of hyperfine shifted resonances of the spectrum have been reported in Dasgupta et al, 2020. Probably this should be clearly stated prior to the description of the current results. In the present version, the peak numbering appears unjustified (where are peaks 1 and 2?, why 10 is missing?...),

while a statement such as "The 1H NMR spectrum of SLAC-T1D has been previously recorded, 18 signals were identified between 60 and 15 ppm (REF)" would help the reader to follow the story. Furthermore, also the fact that the three pairs of resonances 3-5, 9-11, 13-12 arise from exchange process arise from Dasgupta et al (2020), is not explicitly mentioned here.

Line 107: "SLAC-T1D is predominantly in the NI state," should be modified in "The relative intensities of signals in the range 60-22 ppm with those of the region 21-12 ppm shows that SLAC-T1D is predominantly in the NI state"

Line 108: replace "unpaired electrons" with "unpaired electron spin S"

Line 137: "..i.e. the chemical shift increases with an increase in temperature (Bertini et al., 2017, 1993; Bubacco et al., 2000; Tepper et al., 2006)." Here, the reference Bertini et. 2017 might be also replaced with "Banci, L., Bertini, I., and Luchinat, C., The 1H NMR parameters of magnetically coupled dimers -The Fe2S2 proteins as an example, Struct.Bonding, 72, 113-135, 1990"

Figure 2a: In the EXSY/NOESY shown in Figure 2a, some of the cross peaks are apparently quite asymmetric. This is clearly seen for upper- and lower-diagonal cross peaks e2/e1 but also peaks a2/a1 appear unequal. Is this due to the fact that the observed cross peaks (especially e2/e1) are barely detectable from the noise or there is a significant asymmetry in the relaxation properties of these peaks?

Line 178: please replace "in loss of copper or at least severe redistribution of unpaired electron density," with "in loss of copper or at least in a severe redistribution of unpaired electron density,"

Lines 246 and 248: crystal structures 6S0o and 3cg8 should be better called according to the different organism and pdb number.

Lines 287-292: I would not recall Figures 1-4 in the conclusion section. Indeed, these Figures have been extensively discussed throughout the result section; in the final

recap there is no need to go back to the Figures.

Line 293: please replace "the first sequential assignment of the resonance" with "the first sequential assignment of the paramagnetically shifted resonances"

Lines 293-295: I would suggest to somehow smooth the last sentence. Indeed, the blind sphere around the polynuclear copper center might involve many residues besides those that are bound to the T1, T2 and T3 centers. I wonder whether mutagenesis of second shell residues would be enough to provide a full sequence specific assignment.

―――――――――――――――――――――

---

## Short Comment (SC1) · 20 Dec 2020

It is challenging to record 2D NMR spectra and assign resonances of a paramagnetic system of molecular weight > 100 kDa.

To help others to conduct such experiments, would the authors be prepared to deposit their original NMR data in a publicly accessible repository with a doi? This will provide access to parameters of interest, such as the maximal t1-evolution times and carrier frequencies used, as well as giving an impression of the appearance of the spectral regions not shown.

---

## Referee Comment (RC2) · Anonymous Referee #2 · 21 Dec 2020

I agree with the comment made by referee 1 to which I would add the following comments that in my opinion would strengthen the manuscript. Introduction, line 60, none of the references given reports on suggestions that resonance transfer of energy has been suspected or modelled in laccases. Unless there is such expectation and relevant references, this section requires major revision. Chemical exchange in laccases is interesting enough. Line 158: NOE vs Exchange cross peaks have been discriminated in paramagnetic proteins displaying exchange using ROESY experiments. This might offer an opportunity for the authors to obtain conclusive evidence the nature of the cross-peaks. This is relevant in the context of figure 3 where no mention of the assumption is made and the distinction between dipolar and exchange origin of the cross

peaks is taken as firm. Figure 2: The authors should include a 1D spectrum of this region which is not clearly visible in figure 1. It would also be helpful to have together with equation S1 a table of the A and 2J values that were used to generate lines in panel c. Figure 5: it is unclear from the data as reported in the figure how the authors concluded on the existence of crossover for some signals. For example signal 7 in the data from the mutant. I suggest that the figure is made to have the same scale in the two panels.

---

## Referee Comment (RC3) · Anonymous Referee #3 · 21 Dec 2020

Dasgupta et al. present a well-conceived technical study that seeks to resolve the paramagnetic NMR spectrum of the trinuclear copper centre in the small laccase from S. coelicolor. I judge their analysis and interpretation to be sound and correct. The work reported appears to be a follow-on to work reported by the same authors in Biophys. J. 119, 9-14, 2020. To an outsider in this area, it would be useful to have some explicit discussion about how this previous report links into the present work.

Having read referees 1 and 2 reports I agree with their assessment and the changes they have requested, the Table of chemical shift and assignments would be a useful addition. In line with their views and comments I am also positive about this work.

[Figure]

Picking up on referee 2 comments; Abstract, line 11. The authors appear to imply that the efficiency of the kinetically challenging task of reducing molecular oxygen is linked to mobility of active site residues? What is the evidence for this? Is this related to the sentence in the Introduction on line 60? It appears to me that the authors are attempting to correlate dynamics versus catalytic reactivity. Whilst there may well be a case for this, I am not aware that this applies in the present case. The observation of ligand dynamics in the first coordination sphere in a metalloenzyme is interesting, but how does this fit with the entatic state view?

Line 63, characterisation of. . .. Line 102, three pairs of. . .

---

## Author Comment (AC1) · 15 Jan 2021

Reply to the referee comments for the manuscript

Towards resolving the complex paramagnetic NMR spectrum of small laccase: Assignments of resonances to residue specific nuclei

Rubin Dasgupta, Karthick B.S.S. Gupta, Huub J.M. de Groot, Marcellus Ubbink

https://doi.org/10.5194/mr-2020-31

Anonymous Referee #1

1. The manuscript describes the assignment of the paramagnetic NMR spectrum of a small laccase protein, containing a trinuclear copper center. The spectrum has been previously reported, but, In this study, the authors perform the assignment of all hyperfine shifted resonances and succeed in rationalizing the patterns of signals due to chemical exchange processes and to the occurrence of two different redox states: the resting oxidized state and the intermediate state, each of them characterized by peculiar and distinctive features in terms of electron relaxation rates and contact chemical shift ranges. Finally, due to the chemical shift differences observed between the WT protein and Y108F variant, the authors propose the presence of a conformational exchange, involving hydrogen bonds of His 102, which would justify the observed chemical exchange phenomena, and identify at the molecular level the nature of the previously evoked mobility of active site residues. The manuscript is well written, with a very high level of technical skills and a clear description of the results. I recommend its publication, provided the authors addressed a few comments listed below.

Thank you for your kind comments. We have made the following changes based on your suggestions.

2. Also, I would suggest to summarize the assignment of signals as well as the correlations arising from chemical exchange and from the two states, the NI state and the RO state, in a single Table.

We tried to keep it separated as Table S2 and S3 so that the reader does not get confused between the NI and the RO states. We have now mentioned the NI and RO states explicitly in the table heading.

3. Comments: Abstract, line 21. Although it is explained in the introduction section, the expression "T2 histidines" in the abstract is unclear. Please replace it with "histidine residues belonging to the type-2 copper site".

We have changed the phrase to "histidine ligands of the copper in the type-2 site" to make it clear the histidines are copper ligands. There are also non-coordinating

histidines in the neighborhood.

4. Abstract, lines 21-24. The last sentence of the abstract is not clear. Perhaps you might replace it with "This study demonstrates the utility of the approaches used for the sequence specific assignment of the paramagnetic NMR spectra of ligands in the TNC that ultimately may lead to a description of the underlying motion"

Updated the sentence in the abstract

5. line 49, line 69, etc: replace "coppers" with "copper ions" [please spell check throughout the text]

Replaced "coppers" with "copper ions" in the manuscript

6. lines 84-85: same as lines 21-24.

Updated the sentence as comment #4

7. Line 96: The evolution period was shortened to 500 $\mu$s, balancing the time required for formation of antiphase magnetization and paramagnetic relaxation, to optimize S/N ratio for most of the resonances. Actually, previous works on this aspect (such as "I. Gelis, et al; A simple protocol to study blue copper proteins by NMR, Eur. J. Biochem. 270, 600–609, 2003" or "S. Ciofi-Baffoni et al; The IR-15N-HSQC-AP experiment: a new tool for NMR spectroscopy of paramagnetic molecules, Biomol NMR, 58:123-128, 2014") should be quoted here.

We have added the suggested references

8. Lines 97-100: The identification and numbering of hyperfine shifted resonances of the spectrum have been reported in Dasgupta et al, 2020. Probably this should be clearly stated prior to the description of the current results. In the present version, the peak numbering appears unjustified (where are peaks 1 and 2?, why 10 is missing?. . .), while a statement such as "The 1H NMR spectrum of SLAC-T1D has been previously recorded, 18 signals were identified between 60 and 15 ppm (REF)" would

help the reader to follow the story. Furthermore, also the fact that the three pairs of resonances 3-5, 9-11, 13-12 arise from exchange process arise from Dasgupta et al (2020), is not explicitly mentioned here.

This point is made clear by adding the following sentences in section 2.1

The Fermi contact shifted resonances for SLAC-T1D were reported before and here we use the numbering used in our previous study (Dasgupta et al., 2020; Machczynski and Babicz, 2016). Eighteen resonances were found between 15 and 60 ppm. Resonances 1 and 2 were assigned to a region that is attributed to the RO state, therefore we followed the numbering from 3 to 18 in the present work (Figure 1a). Resonance 10 is from a proton bound to carbon and is overlapping with resonances 9 and 11 at temperatures > 293 K (Dasgupta et al., 2020) (Figure 1a). The 1H resonances that exhibited exchange processes (3-5, 9-11 and 13-12) were assigned to Hdelta1 nuclei from histidine coordinated to the copper ion (Dasgupta et al., 2020).

Figure 1a is updated to show the position of resonance 10.

9. Line 107: "SLAC-T1D is predominantly in the NI state," should be modified in "The relative intensities of signals in the range 60-22 ppm with those of the region 21-12 ppm shows that SLAC-T1D is predominantly in the NI state" Line 108: replace "unpaired electrons" with "unpaired electron spin S"

Updated the sentences

10. Line 137: "..i.e. the chemical shift increases with an increase in temperature (Bertini et al., 2017, 1993; Bubacco et al., 2000; Tepper et al., 2006)." Here, the reference Bertini et. 2017 might be also replaced with "Banci, L., Bertini, I., and Luchinat, C., The 1H NMR parameters of magnetically coupled dimers -The Fe2S2 proteins as an example, Struct.Bonding, 72, 113-135, 1990"

Replaced the suggested reference

11. Figure 2a: In the EXSY/NOESY shown in Figure 2a, some of the cross peaks

are apparently quite asymmetric. This is clearly seen for upper- and lower-diagonal cross peaks e2/e1 but also peaks a2/a1 appear unequal. Is this due to the fact that the observed cross peaks (especially e2/e1) are barely detectable from the noise or there is a significant asymmetry in the relaxation properties of these peaks?

There can be a difference in the relaxation properties of these peaks, which, however, requires extensive investigation and is left for future study.

12. Line 178: please replace "in loss of copper or at least severe redistribution of unpaired electron density," with "in loss of copper or at least in a severe redistribution of unpaired electron density,"

Updated the sentence

13. Lines 246 and 248: crystal structures 6s0o and 3cg8 should be better called according to the different organism and pdb number.

We have updated the sentences with

"For example, in the crystal structure of SLAC from Streptomyces griseoflavus, (PDB entry 6s0o resolution 1.8 Å) (Gabdulkhakov et al., 2019) N$\delta$1 of His237 can form a hydrogen bond with Asp114 O$\delta$1 or water O540, depending on rotation around chi2 (Figure S5). In the crystal structure of SLAC from Streptomyces coelicolor (PDB entry 3cg8 resolution 2.68 Å)(Skálová et al., 2009) the equivalent Asp113 O$\delta$1 is moved away from the N$\delta$1 and therefore could not form a hydrogen bond (Figure S5a)."

14. Lines 287-292: I would not recall Figures 1-4 in the conclusion section. Indeed, these Figures have been extensively discussed throughout the result section; in the final recap there is no need to go back to the Figures.

We have removed the figure reference from the conclusion

15. Line 293: please replace "the first sequential assignment of the resonance" with "the first sequential assignment of the paramagnetically shifted resonances"

We have replaced the sentence with

"This report shows the first sequence specific assignment of the paramagnetically shifted resonance to a coordinating histidine"

16. Lines 293-295: I would suggest to somehow smooth the last sentence. Indeed, the blind sphere around the polynuclear copper center might involve many residues besides those that are bound to the T1, T2 and T3 centers. I wonder whether mutagenesis of second shell residues would be enough to provide a full sequence specific assignment.

This study shows that nuclei as close as the H$\delta$1 of histidine ligands can be detected for all eight ligands in the TNC in the NI state. Thus, the blind spot is in fact rather small, only comprising the nuclei at the $\varepsilon$1 and $\delta$2 positions. The idea is that second shell mutants can be applied to identify all eight H$\delta$1 signals and via NOE also H$\beta$ protons in some cases. That yields a multitude of probes to sample motions occurring very near the copper ions.

The final sentence has been changed to: "Clearly, the 'blind spot' due to fast nuclear spin relaxation is small for the TNC in the NI state. Potentially, more second shell residue mutants may help to provide a sequence specific assignment for all histidine ligands, providing a set of probes to study dynamics in the active site and its possible role in the catalytic mechanism."

Anonymous Referee #2

1. I agree with the comment made by referee 1 to which I would add the following comments that in my opinion would strengthen the manuscript.

Thank you for your comments and we have updated the manuscript according to your suggestions.

2. Introduction, line 60, none of the references given reports on suggestions that resonance transfer of energy has been suspected or modelled in laccases. Unless there is

such expectation and relevant references, this section requires major revision. Chemical exchange in laccases is interesting enough.

We have revised this line as

The oxygen reduction process is a multi-step reaction involving transfer of four electron and protons with oxidation and reduction of the copper ions (Scheme 1). Each step is associated with its respective activation energy barrier and the motions of the protein, especially within the active site, may be useful in reduction or crossing of these barriers. Such motions have been reported for many proteins, for example dihydrofolate reductase, adenylate kinase, and cytochrome P450 (Hammes-Schiffer, 2006; Hammes-Schiffer and Benkovic, 2006; Henzler-Wildman et al., 2007; Poulos, 2003).

3. Line 158: NOE vs Exchange cross peaks have been discriminated in paramagnetic proteins displaying exchange using ROESY experiments. This might offer an opportunity for the authors to obtain conclusive evidence the nature of the cross-peaks. This is relevant in the context of figure 3 where no mention of the assumption is made and the distinction between dipolar and exchange origin of the cross peaks is taken as firm.

We agree that the ROESY experiment is useful in discriminating NOE and exchange cross peaks. However, ROESY is expected to be much less sensitive due to the fast transverse relaxation in paramagnetic samples, contributing to T1. Also, the spin-lock can lead to a high duty cycle due to the high repetition rate. The temperature dependence of the cross peaks is also an indicator of exchange peaks. The exchange cross peak integral increases with temperature in the range of 278 to 308 K while the NOE cross peak integral remains unaffected. This was previously shown for SLAC-T1D in our previous work:

Dasgupta, R., Gupta, K. B. S. S., Nami, F., Groot, H. J. M. de, Canters, G. W., Groenen, E. J. J. and Ubbink, M.: Chemical Exchange at the Trinuclear Copper Center of Small Laccase from Streptomyces coelicolor, Biophysical Journal, 119(1), 9–14, https://doi.org/10.1016/j.bpj.2020.05.022, 2020.

Since the EXSY/NOESY spectra of SLAC-T1D and SLAC-T1D/Y108F (Figure 3) are very similar we assumed the cross peaks between pairs 3-5, 9-11 and 13-12 are due to chemical exchange. This is also supported by the similarity in their exchange rates at 298 K shown in Figure S1b and Table S1 of the supporting information.

4. Figure 2: The authors should include a 1D spectrum of this region which is not clearly visible in figure 1.

Panel a showing the 1D 1H WEFT spectrum of the RO state has been added in Figure 2.

5. It would also be helpful to have together with equation S1 a table of the A and 2J values that were used to generate lines in panel c.

We have added the requested table in the supporting information (Table S5) with the values of the hyperfine constant A in MHz. The 2J value and the diamagnetic chemical shift is already mentioned in Figure 2b.

6. Figure 5: it is unclear from the data as reported in the figure how the authors concluded on the existence of crossover for some signals. For example signal 7 in the data from the mutant. I suggest that the figure is made to have the same scale in the two panels.

A new figure is added with the same scale in the y-axis and lines are drawn to the corresponding resonances in the graph itself. The y-axis is the difference between the observed chemical shift and the chemical shift at the lowest temperature which are 278 K and 288 K for SLAC-T1D and SLA-T1D/Y108F respectively. This difference might be same but the chemical shift is different as evident from Figure 1 and Figure 3. This is clarified in the caption of Figure 5 as "SLAC-T1D with temperature relative to 278 K and (b) SLAC-T1D/Y108F with temperature relative to 288 K"

Anonymous Referee #3

1. Dasgupta et al. present a well-conceived technical study that seeks to resolve the

paramagnetic NMR spectrum of the trinuclear copper centre in the small laccase from S. coelicolor. I judge their analysis and interpretation to be sound and correct.

Thank you for your kind comments.

2. The work reported appears to be a follow-on to work reported by the same authors in Biophys. J. 119, 9-14, 2020. To an outsider in this area, it would be useful to have some explicit discussion about how this previous report links into the present work.

We have addressed this as a reply to comment 8 of referee #1.

3. Having read referees 1 and 2 reports I agree with their assessment and the changes they have requested, the Table of chemical shift and assignments would be a useful addition.

The table of chemical shift is given in the supporting information.

4. In line with their views and comments I am also positive about this work.

Thank you for your positive response.

5. Picking up on referee 2 comments; Abstract, line 11. The authors appear to imply that the efficiency of the kinetically challenging task of reducing molecular oxygen is linked to mobility of active site residues? What is the evidence for this? Is this related to the sentence in the Introduction on line 60? It appears to me that the authors are attempting to correlate dynamics versus catalytic reactivity. Whilst there may well be a case for this, I am not aware that this applies in the present case. The observation of ligand dynamics in the first coordination sphere in a metalloenzyme is interesting, but how does this fit with the entatic state view?

The abstract line 11 is revised as "The dynamics of the protein matrix is a determining factor for the efficiency in catalysis"

Line 60 in the introduction is revised and is mentioned as a reply to comment 2 of referee #2. We have deleted the comment about the entatic state and explain that

the multistep reaction of laccase may require different conformations of the enzyme for lowering the respective barriers consecutively.

6. Line 63, characterisation of. . .. Line 102, three pairs of. . .

Corrected the sentences.

Short comment by Prof. Otting

It is challenging to record 2D NMR spectra and assign resonances of a paramagnetic system of molecular weight > 100 kDa. To help others to conduct such experiments, would the authors be prepared to deposit their original NMR data in a publicly accessible repository with a doi? This will provide access to parameters of interest, such as the maximal t1-evolution times and carrier frequencies used, as well as giving an impression of the appearance of the spectral regions not shown.

Thank you for your comments. We have uploaded the raw datasets in zenodo.org with doi: 10.5281/zenodo.4392869